# Assessing the Clinical Effectiveness of Radioimmunotherapy with Combined Radionuclide/Monoclonal Antibody Conjugates in Cancer Treatment: Insights from Randomised Clinical Trials

**DOI:** 10.3390/cancers17091413

**Published:** 2025-04-23

**Authors:** Yifu Chen, Padam Kanta Dahal, Parvez Mosharaf, Md. Shahjalal, Rashidul Alam Mahumud

**Affiliations:** 1School of Public Health, Faculty of Medicine and Health, The University of Sydney, Camperdown, NSW 2006, Australia; yche5552@uni.sydney.edu.au; 2School of Health, Medical and Applied Sciences, Central Queensland University, Sydney Campus, Sydney, NSW 2000, Australia; padamdahal1@gmail.com; 3School of Business, Faculty of Business, Education, Law and Arts, Centre for Health Research, University of Southern Queensland, Toowoomba, QLD 4350, Australia; parvez.mosharaf@unisq.edu.au; 4Bioinformatics Laboratory, Department of Statistics, University of Rajshahi, Rajshahi 6205, Bangladesh; 5Department of Public Health, North South University, Dhaka 1212, Bangladesh; shahjalalsiam@gmail.com; 6NHRMC Clinical Trials Centre, Faculty of Medicine and Health, The University of Sydney, Camperdown, NSW 2006, Australia

**Keywords:** cancer, non-Hodgkin’s lymphoma, radioimmunotherapy, systematic review and meta-analysis

## Abstract

This is the first meta-analysis to assess the effectiveness of radioimmunotherapy (RIT) for various cancers by including only randomised controlled trial study designs. The inclusion of 20 clinical trials with a total of 3562 patients assigned to 10 different RITs covers the last 20 years of RIT research in seven types of cancer (including non-solid and solid tumours). Our analysis presents a comprehensive evaluation of the efficacy and safety profiles of various cancer treatments based on a synthesis of direct and indirect evidence derived from the included trials and an investigation of different patient subgroups. Our analysis provided an in-depth focus on the most commonly used RIT, the most common cancer types treated by RIT, and the patient population that has been treated with RIT. This allowed us to contextualise and identify potential directions for optimising cancer treatment regimens and future RIT development.

## 1. Introduction

Despite the existence of various advanced cancer therapies, the eradication of cancer remains a challenging issue in the field of oncology. Radioimmunotherapy (RIT) is an innovative therapeutic approach that combines radionuclides with monoclonal antibodies (mAbs) directed against tumour-associated antigens or antigens expressed by the cells of the tumour microenvironment [1].

RIT has been extensively researched and trialled over the past two decades [2]. To date, only two RITs have been approved by the U.S. Food and Drug Administration (FDA) for the treatment of non-Hodgkin’s lymphoma (NHL) [2,3]. For instance, the radiolabelled anti-CD20 mAbs ^90^Y-labelled ibritumomab tiuxetan (Zevalin^®^; Seattle Cellular Therapeutics, San Diego, CA, USA) and the ^131^I-labelled tositumomab (Bexxar^®^; GlaxoSmithKline LLC, Seattle, WA, USA) were approved in 2002 and 2003, respectively. However, due to low utilisation, the latter RIT has been withdrawn from the market [1].

Evidence shows that RIT is an effective treatment for non-solid tumours, particularly NHL, but its clinical use is relatively limited [2,3]. In addition, since most clinical trials in solid tumours have not progressed to phase III trials, and the clinical trial data of the only RIT approved by the China State FDA (i.e., metuximab) for solid tumour have been disappointing, the efficacy of RITs for solid tumours is still inconclusive, and the clinical evidence is insufficient [2]. Nevertheless, current interest in RIT research is shifting from non-solid tumours to solid tumours, as evidenced by the significant decrease in the number of clinical trials of RITs for non-solid tumours compared to solid tumours over the past decade [2]. In addition to considering the efficacy of RITs from a clinical application perspective, improving the efficacy of RITs also requires the combined efforts of multidisciplinary practitioners, and addressing the radiation safety issues associated with RITs therefore requires consideration of the economics of producing radiolabelled compounds and isotopes [3,4]. As such, studies with comprehensive evaluation to understand the overall efficacy of RIT in clinical use are limited [1,2,3,5,6,7,8].

The burden of cancer is a continuing concern, not only in the field of oncology, but also in the healthcare system. A comprehensive understanding of innovative cancer therapies is not only essential for the development of effective cancer treatment strategies but also facilitates the optimisation of treatment outcomes for cancer patients. Therefore, the primary objective of this systematic review and meta-analysis is to analyse the existing clinical trials to investigate the overall effectiveness of RIT for cancer treatment and how RIT may influence treatment outcomes in cancer patients. This review therefore fills this gap, focusing on clinical trials using only the RCT study design and providing a comprehensive analysis of all cancer types treated with RIT.

## 2. Methods

### 2.1. Design and Registration

This review adhered to the Preferred Reporting Items for Systematic Review and Meta-analyses (PRISMA) guidelines [9] and was structured and designed in accordance with the PICOS framework [10].

-Population (P): All patients receiving RIT for cancer treatment.-Interventions (I): RIT for cancer treatment.-Comparison (C): Conventional and/or emerging cancer therapies.-Outcomes (O): Treatment outcomes (e.g., overall survival, disease-free survival, progression-free survival) in cancer patients treated with RIT.-Study Design (S): This review focuses on RCTs to evaluate the overall effectiveness of RIT.

This review is registered in PROSPERO (Registration number: CRD42024584616).

### 2.2. Eligibility Criteria

In this review, the inclusion and exclusion criteria were determined in accordance with the research objectives, the study population, the interventions, the comparisons, and the results (i.e., the PICOS framework) [10].

The inclusion criteria for this study are as follows: studies involving patients who received RIT for cancer treatment; clinical trials using an RCT design; studies that stated the cancer type, cancer stage, and patient demographics; studies that compared RIT with other cancer therapies; studies that provided sufficient data for the synthesis of treatment outcomes; and studies published in English between January 2000 and October 2024, with the objective of capturing the most pertinent and up-to-date articles on this subject. However, studies that do not contain specific outcomes, studies with insufficient data, reviews, editorials, reports, in vitro trials, in vivo trials, grey literature, commentaries, qualitative studies, and perspectives were excluded from this review.

### 2.3. Search Strategy, Study Screening, and Selection

A search strategy was developed to search PubMed, EMBASE, Scopus, the Cochrane Library (CENTRAL), and Google Scholar to identify relevant clinical trials for this review. In addition to the database searches, references of reviews on similar topics were examined, and manual searches of other resources such as ResearchGate and Clinicaltrials.gov were conducted to incorporate all potential relevant clinical trials.

The search strategy contained four concepts: (1) whether RIT is used in combination with other cancer therapies or alone; (2) treatment outcomes of RIT; (3) adverse events associated with RIT; and (4) other conventional and emerging cancer therapies. The formulation of search terms was achieved through the utilisation of a combination of keywords, the Medical Subject Headings (MeSH), and Boolean operators. A comprehensive account of the search terms utilised is presented in Appendix A.

The authors conceptualised this study, reviewed the search strategies, and developed the protocol. Then, the articles retrieved from the database searches were exported in EndNote 20.6 (Clarivate^TM^, Philadelphia, PA, USA), duplicates were removed, and the title and abstract were screened independently by two authors. Full-text screening of the selected articles was performed by two authors and confirmed with co-authors. In the final stage, data extraction and quality assessment were conducted in consultation with co-authors.

### 2.4. Data Extractions

A data extraction sheet was developed based on the PRISMA guidelines. The data were then extracted using Microsoft Excel worksheets. The recorded relevant data were the title of the included study, name of the first author, study setting (country), year of publication, sample size, intervention and control, time frame/follow-up period, patient population (e.g., median age/age group, gender, pre-treatment disease status), cancer type and stage, RIT (including targeted antigen, antibody, and isotope), treatment regimen, other cancer therapies, adverse events related to RIT, and the treatment outcome. Then, further recorded data on treatment outcomes (i.e., length of survival, survival rates, hazard ratio) were categorised and analysed.

### 2.5. Quality Assessment

The Joanna Briggs Institute (JBI) Critical Appraisal Tool for RCTs was employed to evaluate the potential for bias in the included clinical trials [11]. The checklist consists of 13 items, which examine internal validity (including bias related to selection, allocation, assessment, detection, measurement of the outcome, participant retention) and statistical conclusion validity [11]. Each item has four options with different scores: ‘Yes’ (1), ‘No’ (0), ‘Not clear’ (0.5), and ‘Not applicable’. Based on the final score of the study, the tool provides a subjective assessment of the risk of bias (classified as low, medium, or high).

### 2.6. Data Synthesis and Analysis

The present review employed a narrative synthesis approach to examine the interconnections between the included studies and to offer a comprehensive evaluation of the evidence for the effectiveness of RIT in clinical trials [12]. This approach allows for a thorough examination of the influence of RIT on the results of diverse cancer therapies and its role in clinical trials, its relationship with other cancer treatments, and the impact of its safety and toxicity on cancer treatment outcomes.

Furthermore, in meta-analyses, overall pooled hazard ratios for disease progression and overall survival were calculated. Statistical heterogeneity was tested using the I^2^ statistic, with I^2^ > 50% considered high heterogeneity. In such cases, the study design and characteristics of the included studies were further analysed. Similarly, if statistical heterogeneity was observed (I^2^ ≥ 50%), the random-effects model for meta-analysis was performed. Each outcome was combined, calculated using the statistical software STATA/SE 15, and forest plots were used to show the distribution of effect estimates.

## 3. Results

### 3.1. Description of Studies Included

A total of 2241 records were identified by searching the electronic databases mentioned in the Section 2. In our initial search, a total of 25 trials were identified and are summarised. However, upon applying our inclusion/exclusion criteria and the quality assessment (using the Joanna Briggs Institute (JBI) Critical Appraisal Tool), only 20 studies met all the criteria for inclusion in the systematic analysis. Ultimately, 20 [13,14,15,16,17,18,19,20,21,22,23,24,25,26,27,28,29,30,31,32] clinical trials were deemed suitable for inclusion in this review (Figure 1).

### 3.2. Characteristics of Included Clinical Trials

Table 1 shows the characteristics of the studies included in this review. A total of 40% (*n* = 8) studies were conducted in the United States (USA) [13,14,22,23,25,29,30,32], while 20% (*n* = 4) were carried out in China [17,24,26,27]. Most of the included studies (approximately 55%, *n* = 11) were phase III clinical trials, while 25% (*n* = 5) were phase II clinical trials.

Of the clinical trials included in this review, approximately 60% (*n* = 12) focused on non-solid tumours [13,14,18,19,21,22,23,25,28,30,31,32]. Among them, 10 studies investigated the treatment for non-Hodgkin’s lymphoma (NHL), and 1 focused on the treatment for acute myeloid leukaemia (AML) [30]. Among the studies on non-solid tumours, eight (40%) focused on low-grade NHL including follicular lymphoma (FL) [14,18,19,22,25,28,31,32], and two (10%) focused on high-grade NHL including diffuse large B-cell lymphoma (DLBCL) [13,23].

Eight (40%) studies focused on solid tumours, of which four (20%) focused on lung cancer [17,24,26,27], followed by pancreatic adenocarcinoma [20], prostate cancer [29], gliomas [16], and epithelial ovarian cancer (EOC) [15].

^90^Y-ibritumomab tiuxetan was used in seven (35%) studies [13,18,19,21,28,31,32], followed by ^131^I-tositumomab in four (25%) studies [14,22,23,25] and ^131^I-metuximab in three (12.5%) trials [17,24,27]. The remaining RITs used ^131^I-apamistamab [30], ^90^Y-human milk fat globule 1 murine monoclonal antibody (^90^Y-muHMFG1) [15], ^131^I-anti-CEA monoclonal antibody (^131^I-Kab201) [20], ^125^I-monoclonal antibody 425 (^125^I-MAb425) [16], ^131^I-mouse/human chimeric monoclonal antibody (^131^I-chTNT) [26], and ^177^Lu-anti-prostate-specific membrane antigen (^177^Lu-J591) [29].

Eleven (55%) studies [13,14,18,19,21,22,23,25,28,31,32] used CD20 as a target, followed by three (15%) studies [17,24,27] that used CD147/HAb18G as a target. The remaining targets were MUC1, EGFR, CEA, DNA/Histone H1, PSMA, and CD45, each of which was used once in the remaining studies, respectively.

### 3.3. Reported Treatment Outcomes

Table 2 shows the treatment outcomes and key findings of the clinical trials. Overall survival (OS) data were reported in 13 (65%) studies [16,17,20,21,22,23,24,25,26,27,28,30,31]. Among them, eight (61.5%) of these showed that the RIT group had a relatively higher OS (survival rate or length of time) compared to the comparator group [17,21,23,24,26,27,30,31]. 

Progression-free survival (PFS) was reported in 11 (55%) studies [14,18,19,21,22,23,25,28,29,31,32], where 8 studies showed that the RIT group had a relatively higher PFS (survival rate or length of time) compared to the comparator group [14,18,19,22,23,29,31,32]. Event-free survival (EFS) was reported in four (25%) studies [25,28,30,31], of which two studies showed that the RIT group had a relatively higher EFS (survival rate or length of time) compared to the comparator group [30,31].

Thirteen (65%) studies concluded that RIT is effective in improving treatment outcomes [13,14,17,18,19,22,24,25,26,27,30,31,32], with these trials treating cancers including NHL, HCC, FL, AML, and NSCLC. However, three (15%) of the studies concluded that RIT is not effective in improving treatment outcomes [15,16,23]. Similarly, four studies (20%) were inconclusive on the effectiveness of RIT [20,21,28,29].

### 3.4. Comparison of RIT Therapy with Other Therapies

The comparative analysis in Table 3 reveals that radioimmunotherapy (RIT) is predominantly integrated as part of combination regimens rather than deployed as monotherapy. Four (20%) studies compared RIT combination therapy with rituximab alone [13,25,28,32], of which three studies demonstrated that the RIT combination therapy yielded superior outcomes in comparison to rituximab monotherapy [13,25,32]. Six studies used RIT combination therapy as consolidation therapy [15,18,19,22,28,31]. One study used RIT monotherapy [14]. Three studies used RIT treatment with transplant therapy [17,21,30]. The remaining six studies used RIT treatment with various special therapies specific to the cancer being treated, including teleradiotherapy, chemotherapy, radiofrequency ablation (RFA), percutaneous coagulation therapy (PMCT), hepatectomy, and ketoconazole [16,23,24,26,27,29].

In several non-Hodgkin’s lymphoma trials, RIT was combined with established agents such as rituximab, with studies by Witzig et al. (2002) [13], Morschhauser et al. (2008) [18], Goff et al. (2009) [19], and Laoruangroj et al. (2024) [32] demonstrating improved outcomes over rituximab alone. Similarly, in trials evaluating consolidation and adjuvant therapies, such as those by Shimoni et al. (2012) [21] and Press et al. (2013) [22], the addition of RIT to regimens including BEAM chemotherapy, ASCT, or CHOP-based treatments yielded enhanced clinical benefits. Moreover, studies on solid tumours indicated that integrating RIT with other modalities—ranging from surgical interventions and radiotherapy to innovative approaches like PMCT and alloHCT—may optimise treatment responses. Overall, these findings suggest that RIT, when combined with standard or emerging therapies, holds promise for improving treatment efficacy across various cancer types, while underscoring the need for further research to determine the optimal therapeutic combinations and patient selection strategies.

Table 3 summarises findings from various clinical trials evaluating the effectiveness of radioimmunotherapy (RIT) in combination with other treatments such as chemotherapy, immunotherapy, surgery, and targeted therapy across multiple cancer types. Notably, RIT combination therapy consistently demonstrated superiority over several comparator treatments, including rituximab alone, placebo, standard chemotherapy regimens (e.g., BEAM, R-BEAM), radiofrequency ablation (RFA), salvage therapies, autologous stem cell transplantation (ASCT), and multimodal approaches involving surgery, chemotherapy, and radiotherapy (Witzig et al., 2002; Xu et al., 2007; Shimoni et al., 2012; Bian et al., 2014; Zhao et al., 2016; Gyurkocza et al., 2024; Ladetto et al., 2024) [13,17,21,24,26,30,31]. Conversely, outcomes from RIT combinations were comparable or showed no significant difference compared to standard chemotherapy alone, R-CHOP, or rituximab monotherapy in other trials (Verheijen et al., 2006; Press et al., 2013; Vose et al., 2013; López-Guillermo et al., 2022; Laoruangroj et al., 2024) [15,22,23,28,32]. However, caution is warranted, particularly in solid tumours, as the data from Tagawa et al. (2023) [29] indicated prolonged survival with RIT combinations but at the expense of increased toxicity. Thus, while RIT combination therapies appear promising and consistently beneficial for haematological malignancies, their utility in solid tumours remains uncertain, highlighting the need for further investigation into targeted antigen selection and patient stratification to optimise clinical benefits.

#### 3.4.1. Forest Plot for Progression-Related Outcomes

The forest plot for progression-related endpoints shows individual hazard ratios (HRs) for each study along with their corresponding 95% confidence intervals (CIs) (Figure 2). Nearly all studies report HRs below 1.0, indicating that the RIT interventions are associated with a reduction in the risk of progression or time to progression compared to controls. The pooled HR from the random-effects model was approximately 0.48 (95% CI: 0.39–0.59), suggesting that these treatments reduce the risk of progression by roughly 52%. This finding underscores the potential of multi-outcome evaluations to capture meaningful improvements in disease control.

#### 3.4.2. Forest Plot for Overall Survival (OS)

In contrast, the forest plot summarising OS outcomes displays more variability (Figure 3). Several studies report HRs favouring the intervention, while others show estimates close to or even exceeding 1. The overall pooled OS HR is around 0.80 (95% CI: 0.60–1.07), indicating a trend toward improved survival; however, the confidence interval crosses the null, and statistical significance is not achieved. The greater heterogeneity (I^2^ > 50%) observed in OS estimates may reflect differences in follow-up durations, subsequent lines of therapy, or variations in patient populations across studies. This variability suggests that while RIT may consistently delay disease progression, the overall survival benefit is less consistent, possibly due to the influence of post-progression treatments and differences in trial design.

### 3.5. Safety and Toxicity

Table 4 summarised the adverse events (AEs) reported in the clinical trials. Except for the six clinical trials (30%) that either did not report AEs or did not specify the assessment tool used to categorise AEs, almost all of the remaining trials (*n* = 13, 65%) used the National Cancer Institute’s Common Terminology Criteria for Adverse Events (CTCAE) or Common Toxicity Criteria (CTC) for assessment. According to CTCAE versions 3.0 and 4.0, lymphopenia was defined as <500–200/mm^3^ (grade 3) and <200/mm^3^ (grade 4), thrombocytopenia (low platelet count) was defined as <50,000–25,000/mm^3^ (grade 3) and <25,000/mm^3^ (grade 4), neutropenia (low neutrophil count) was defined as <1000–500/mm^3^ (grade 3) and <500/mm^3^ (grade 4), and leukopenia (low leukocyte counts) was defined as <2000–1000/mm^3^ (grade 3) and <1000/mm^3^ (grade 4). Anaemia (low haemoglobin level) was defined as <8.0–6.5 g/dL (grade 3) and <6.5 g/dL (grade 4) in version 3.0, whereas it was defined as <8.0 g/dL (grade 3) and life-threatening (grade 4) in version 4.0. According to CTC version 2.0, lymphopenia was defined as <500/mm^3^ (grade 3), thrombocytopenia was defined as ≥10,000–<50,000/mm^3^ (grade 3) and <10,000/mm^3^ (grade 4), neutropenia was defined as ≥500–<1000/mm^3^ (grade 3) and <500/mm^3^ (grade 4), and leukopenia was defined as ≥1000–<2000/mm^3^ (grade 3) and <1000/mm^3^ (grade 4).

A total of 18 clinical trials (90%) reported AEs; among them, 17 (85%) studies reported haematological AEs [13,14,15,18,20,21,22,23,24,25,26,27,28,29,30,31,32]. Further, 16 (80%) studies reported nonhaematological AEs [13,14,15,18,21,22,23,24,25,26,27,28,29,30,31,32] including nausea, vomiting, pain, fatigue, diarrhoea, anorexia, infection, and second neoplasm. In addition, AEs such as increased alanine aminotransferase (ALT), aspartate aminotransferase (AST), and alkaline phosphatase (ALP) were also reported in three studies [24,27,29].

### 3.6. Quality of Included Studies

Seven (35%) studies were considered to be of high quality [13,14,17,19,24,27,29], and the remaining 13 (65%) of the included clinical trials were considered to be of medium quality [15,16,18,20,21,22,23,25,26,28,30,31,32] (Appendix A).

## 4. Discussion

This review assessed the effectiveness of RIT for cancer treatment and its influence on treatment outcomes among cancer patients. Furthermore, it analysed data from 20 clinical trials involving approximately 3000 cancer patients and summarised the treated cancer types, key findings of the trials, association of RIT with other therapies, and adverse events associated with RIT. Overall, the meta-analysis figures highlight that RIT interventions consistently improve progression-related outcomes, while their effects on overall survival are more variable.

As expected, most of the studies included in this review were conducted on non-solid tumours, with only a few of them progressing to phase III trials in solid tumours, which is similar to the findings reported by Rondon et al. in 2022 [2]. Further, most of these studies pertained to NHL, which indicates the possibilities of using RIT for the treatment of non-solid tumours. This finding is consistent with previous reviews which indicated that RIT is effective against indolent/low-grade NHL, especially for R/R NHL [1,2,3,6]. Similarly, our review found that the RITs used for non-solid tumours were mainly ^90^Y-ibritumomab tiuxetan and ^131^I-tositumomab, which are the only RITs approved by the FDA for the treatment of NHL [1,2,3,5,6,38,39]. The two approved RITs both target the CD20 antigen, while the RIT used in AML, ^131^I-apamistamab, targets the CD45 antigen. This suggests that research targeting antigens for non-solid tumours should not focus solely on CD20 and that broadening the scope of target antigen research may enable the development of other potential RITs for the treatment of non-solid tumours, which is supported by a previous review [6]. Attempts to use RIT in solid tumour types are broader, varying from lung cancer, EOC, pancreatic carcinoma, to prostate cancer. One of the lung cancer trials included is for NSCLC, which used a different RIT, ^131^I-chTNT, which has also been reported in the treatment of brain tumours [40,41,42]. This indicates the possibility of RIT to be used to treat different cancers. The antigen targeted by ^131^I-metuximab, CD147, showed positive results in three liver cancer trials, demonstrating that CD147-targeted RIT was effective in treating liver cancer; further research in liver cancer may continue to focus on the CD147 antigen. However, other tumour-associated antigens for solid tumours did not show encouraging results. MUC1 was used as a target for the treatment of epithelial ovarian tumour in the included trial and for pancreatic tumour in another trial [43], but this trial was terminated prematurely without a positive outcome for RIT. EGFR was used as a target for the treatment of gliomas in the included trial; while Wygoda et al. [16] concluded that EGFR-targeted RIT was not effective in high-grade gliomas, another phase II trial showed encouraging results in glioblastoma multiforme [44]. However, a large RCT study is still needed to confirm its effectiveness in the treatment of brain tumours. PSMA has been used as a main target for treating prostate cancer, and it has shown encouraging results when used as radioligand therapy (i.e., ^177^Lu-PSMA-617 [45]); however, Tagawa et al. [29] concluded that high toxicity followed treatment when it is used as RIT even though it prolonged survival. This highlights the need to further improve the safety profile of PSMA-targeted RIT for translation to the clinic. Regarding CEA, it was used as a target for treating pancreatic cancer in the included trial. While the trial did not provide a definitive conclusion regarding the effectiveness of RIT compared to conventional chemotherapy, RIT did prolong survival. Therefore, it can be assumed that using CEA as a target for treating pancreatic cancer is a promising direction. Unlike antigens found either in the cell membrane or inside the cell, DNA/histone H1 is a specific target found in the nucleus and was also used in one of the included trials. This provides the possibility of investigating RIT using monoclonal antibodies that target components in the nucleus.

In addition to the antigen, the radionuclides used were also important in determining the efficacy of RIT. Most of the included trials used yttrium-90 (Y-90) and iodine-131 (I-130), with two trials using iodine-125 (I-125) and lutetium-177 (Lu-177), respectively. Importantly, all of the included trials using Y-90 were for the treatment of lymphoma, whereas the trials using I-130 were also for the treatment of solid tumours, mainly liver cancer, as well as lymphoma. I-125 and Lu-177 were both used exclusively for the treatment of solid tumours in the present study. Y-90, I-130, and Lu-177 all have a relatively short half-life of about 2–8 days, while I-125 has a longer half-life of about 59.4 days. Interestingly, it is said that the properties of I-125 make it a preferred isotope for radiotherapy in the treatment of brain tumours and prostate cancer, but the prostate cancer treated in this study used Lu-177 instead [46]. Furthermore, contrary to the suggestion by Leonard et al. [47] that Y-90 would be preferable to 131-I for patients with solid tumours, 131-I has been used extensively for the treatment of liver cancer, with encouraging results in all cases. This highlights that 131-I would be useful for the treatment of solid tumours. In addition, the more encouraging results for lymphoma in the studies using Y-90 rather than 131I suggest that Y-90 may be more suitable for the treatment of non-solid tumours. In general, the choice of a suitable radionuclide is important for the clinical treatment effect of RIT, but not only the chemical properties themselves, but also the availability, cost-effectiveness, and safety profile should be considered.

Previous reviews have reported that RIT is effective for NHL, demonstrating improved survival rates, high success rates, excellent anti-lymphoma effects, and the ability to maintain long-term remission and produce long-term responses in patients [1,2,3,5,6,38]. These findings are analogous to those of the present review, which demonstrate that the treatment outcome indicators (e.g., OS, PFS, EFS, TTP, etc.) reported in most NHL trials indicate that the treatment outcomes of the RIT group were superior to those of the control group.

It is also noteworthy that most non-solid tumour trials (5 out of 12) included patients who had previously undergone treatment and had experienced a recurrence or relapse [13,14,21,25,31]. This indicates that RIT is frequently utilised as an alternative therapeutic modality for patients lacking other treatment options, particularly in the context of NHL. In contrast, the majority of trials conducted on solid tumours (four out of eight) either failed to corroborate the efficacy of RIT or yielded inconclusive results regarding its effectiveness [15,16,20,29]. These findings are in accordance with the results of previous reviews, which indicate that RIT is less effective in the treatment of solid tumours [1,2,3,38,39]. Nevertheless, further, larger-scale trials are required to substantiate the efficacy of RIT in the treatment of HCC.

Our review identified that RIT is employed almost exclusively as an adjuvant therapy in cancer treatment regimens, rather than as a standalone therapy. These trials used RIT in combination with rituximab, chemotherapy, transplantation, and other therapies (e.g., radiotherapy, surgery, etc.). This is in line with previous reviews suggesting that RIT is effective when used in combination with other therapies to enhance efficacy [1,3,38]. Conversely, RIT combination therapy has demonstrated efficacy when it is employed in conjunction with rituximab, as an adjuvant therapy for patients requiring transplantation, or in combination with other tailored therapies in select circumstances (e.g., specific cancers).

In terms of the safety and toxicity of RIT, almost all of the trials (18 out of 20) identified AEs associated with RIT. Most of the reported AEs are haematological and nonhaematological. The haematological AEs were classified as grade 3/4 haematological AEs, encompassing lymphopenia, thrombocytopenia, (febrile) neutropenia, anaemia and leukopenia. This is consistent with the findings of previous reviews indicating that myelosuppression is the main AE of RIT; however, it is reversible, with patients recovering after treatment [3,5]. Similarly, approximately 35% of the selected studies reported a higher incidence of nonhaematological AEs in the RIT group. Common nonhaematological AEs reported in the included trials were of moderate intensity and included nausea, vomiting, pain, fatigue, diarrhoea, and infection. These occurred in 16 out of the 18 trials in this review. While drug-related mortality was rare in the included trials, the presence of these AEs should be considered and managed with caution when using RIT to treat cancer patients, particularly those who are frail or elderly. In addition, when using RIT as a treatment option for cancer, the patient’s condition and response to medication should always be closely monitored, and certain medications for severe AE should be made available when necessary. In addition to these approaches to managing AEs, it is more important to address the underlying causes of AEs by discovering safer radionuclides and developing new types of RITs while maintaining the same therapeutic efficacy.

Most of the included trials were of moderate quality due to open-label designs, unclear randomisation processes, and limited internal validity of outcome measures. Further, most of the high-quality trials primarily focused on solid tumours, including lung cancer, NHL, prostate cancer, and pancreatic cancer.

### 4.1. Strengths and Limitations

This manuscript offers several notable strengths that enhance its contribution to the field of cancer therapy. First, by focusing exclusively on randomised controlled trials, this study provides a robust and high-quality synthesis of the available evidence on radioimmunotherapy (RIT). The inclusion of 20 clinical trials spanning over 20 years and covering seven distinct cancer types, including both non-solid and solid tumours, underscores the comprehensive nature of the analysis. Such breadth not only strengthens the validity of the findings but also provides a rich dataset for examining both efficacy and safety profiles of various RIT modalities.

A key strength of this work is its dual focus on treatment outcomes and adverse event profiles. By evaluating endpoints such as overall survival, progression-free survival, and event-free survival alongside detailed reporting of both haematological and nonhaematological adverse events, this study offers a nuanced view of the clinical trade-offs associated with RIT. This approach is particularly valuable given the complex interplay between treatment efficacy and toxicity in cancer care. Moreover, this manuscript highlights the frequent use of RIT as an adjuvant rather than as a standalone therapy, which reflects current clinical practice and points to the potential for synergistic effects when RIT is combined with other treatment modalities.

This review has several limitations. The principal limitation of this study is that the impact of RIT as a standalone treatment on cancer therapy outcomes could not be evaluated, as most included studies did not utilise RIT monotherapy. This review exclusively considered survival indicators (e.g., OS, PFS, etc.), and did not include indicators such as tumour response, which is another crucial factor when evaluating the efficacy of cancer treatments. Due to the inherent complexity of incorporating all types of indicators for analysis, some relevant additional indicators have been excluded from this study. Consequently, the actual role of RIT in cancer treatment may not be fully reflected. Fourthly, the restrictions on the study design meant that some important trials that did not meet the requirements (such as single-arm trials with a large sample size) were excluded, which may have reduced the representativeness of the included trials. In addition, this review did not encompass other outcome measures beyond health-related indicators, such as the quality of life of patients and the economic outlook (for the healthcare system or for patients), which are also crucial elements when evaluating the efficacy of novel cancer treatments.

### 4.2. Implications of Findings

The implications of these findings are multifold. Clinically, the data emphasise the effectiveness of RIT in non-solid tumours, particularly in cases of non-Hodgkin’s lymphoma, where targeting the CD20 antigen has yielded favourable outcomes. This suggests that for patients with limited treatment options, RIT represents a viable alternative or complementary approach. However, the relatively inconclusive results in solid tumour trials indicate a critical need for further investigation, possibly through larger-scale and more targeted studies. The apparent narrow focus on CD20 as a target in non-solid tumours also suggests that expanding research to encompass additional tumour-associated antigens could extend the therapeutic applicability of RIT.

From a research and policy perspective, this review underscores the importance of multi-modality treatment strategies. The promising outcomes associated with RIT in combination with other therapies, such as rituximab, chemotherapy, and transplantation protocols, open avenues for optimising treatment regimens. Additionally, while adverse events were common, their generally reversible nature provides a framework for improving safety management and guiding patient selection criteria. To further enhance clinical relevance, we have included additional recommendations for managing toxicities in practice. These strategies include the following: routine complete blood count (CBC) monitoring and timely dose adjustments for haematological toxicities; prophylactic and/or rescue use of growth factors (e.g., G-CSF) to mitigate neutropenia; administration of antiemetics to prevent or control nausea and vomiting; proactive fluid and electrolyte management in patients at risk of dehydration or renal complications; careful assessment of the need for blood product transfusions in cases of severe anaemia or thrombocytopenia; and close collaboration among oncologists, pharmacists, and supportive care teams to ensure individualised management for older adults or those with comorbidities. These steps, coupled with ongoing research into safer radionuclides and novel RIT approaches, are essential for balancing treatment efficacy with optimal patient safety and comfort. Future studies should aim to refine dosing strategies and explore novel radiolabelling techniques to mitigate toxicity.

## 5. Conclusions

This review concluded that RIT is indeed effective against non-solid tumours and therefore may be a useful alternative for patients with limited treatment options. Nevertheless, despite the favourable outcomes observed in the included lung cancer trials, further investigation is necessary to ascertain the efficacy of RIT against solid tumours. Most non-solid trials included in this review were focused on the CD20 antigen target, which may impede the development of RIT for the treatment of other cancers besides NHL. Despite the relatively encouraging results of combined RIT therapy, RIT is almost exclusively used as an adjuvant therapy in cancer treatment regimens, rather than as a standalone therapy. Both haematological and nonhaematological AEs occur frequently in the included trials, underscoring the necessity for further development of RIT to mitigate its associated toxicity, as well as the need for improved management of AEs and cautious consideration when applying RIT to specific patient populations.

## Figures and Tables

**Figure 1 cancers-17-01413-f001:**
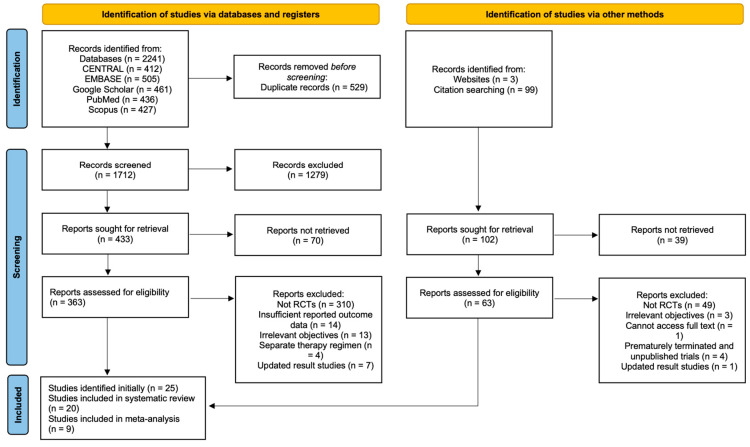
PRISMA flow diagram.

**Figure 2 cancers-17-01413-f002:**
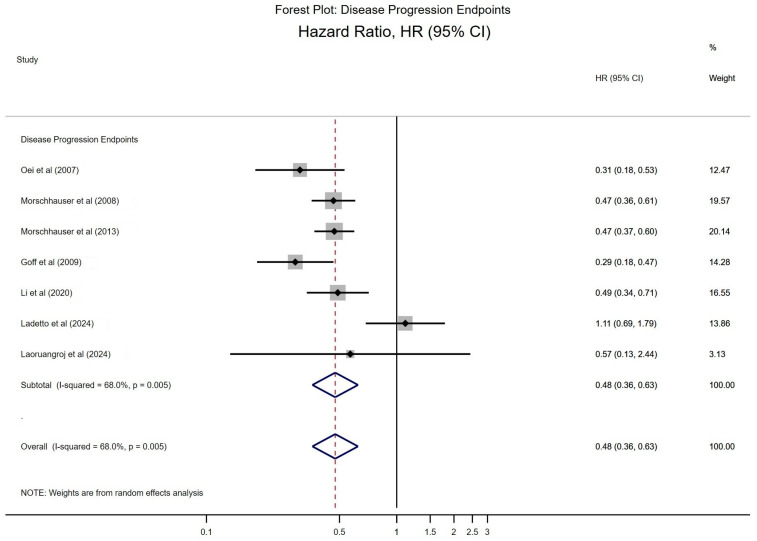
Meta-analysis of disease progression regarding the effectiveness of radioimmunotherapy [8,18,19,25,31,32,36].

**Figure 3 cancers-17-01413-f003:**
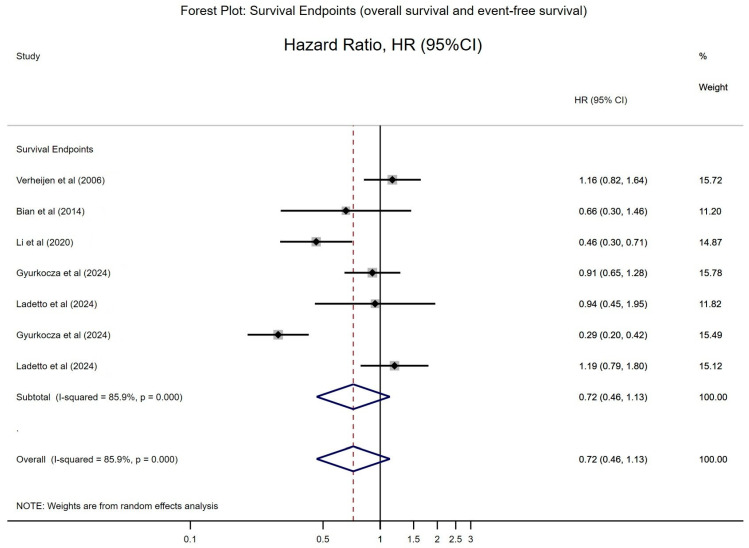
Meta-analysis of survival endpoints regarding the effectiveness of radioimmunotherapy [16,24,27,30,31].

**Table 1 cancers-17-01413-t001:** Characteristics of included clinical trials.

Study (Year)	Country	Sample Size (I/C) (n)	Gender Age (I/C)	Clinical Phase	Cancer Treated (Type)	Cancer Type	RIT Used	Antigen Targeted
Witzig et al. (2002) [13]	USA	143 (73/70)	Both genders Median: 60/57	III	NHL (Stage III/IV)	Non-solid	^90^Y-ibritumomab tiuxetan	CD20
Davis et al. (2004) [14]	USA	78 (42/36)	Both genders Median: 56/55	II	NHL (Low-grade or transformed low-grade)	Non-solid	^131^I-tositumomab	CD20
Verheijen et al. (2006) [15]	The Netherlands	447 (224/223)	Female Median: 54.5/53.7	III	EOC (Stage ≥ Ic ^a^)	Solid	^90^Y-muHMFG1	MUC1
Wygoda et al. (2006) [16]	Poland	18 (8/10)	Both genders Median: 47/52	N/A	Gliomas (Grade III/IV)	Solid	^125^I-MAb425	EGFR
Xu et al. (2007) [17]	China	60 (30/30)	Both genders Median: 44.5/42.5	N/A	HCC (Stage III/IV)	Solid	^131^I-metuximab	CD147/HAb18G
Morschhauser et al. (2008) [18]	Germany	414 (208/206)	Both genders Median: 55/53	III	FL (Stage III/IV)	Non-solid	^90^Y-ibritumomab tiuxetan	CD20
Goff et al. (2009) [19]	United Kingdom	186 (90/96)	Both genders Median: 54/51	III	FL (Stage III/IV)	Non-solid	^90^Y-ibritumomab tiuxetan	CD20
Sultana et al. (2009) [20]	United Kingdom	19 (10/9)	Both genders Median: 59/60	I/II	Pancreatic adenocarcinoma (Stage IVa/IVb)	Solid	^131^I-Kab201	CEA
Shimoni et al. (2012) [21]	Israel	43 (22/21)	Both genders Median: 58/51	III	NHL (High-grade)	Non-solid	^90^Y-ibritumomab tiuxetan	CD20
Press et al. (2013) [22]	USA	532(265/267)	Both genders Median: 53.3/54.5	III	FL (Stage II, III, or IV)	Non-solid	^131^I-tositumomab	CD20
Vose et al. (2013) [23]	USA	224 (111/113)	Both genders Median: 56.8/58.5	III	DLBCL	Non-solid	^131^I-tositumomab	CD20
Bian et al. (2014) [24]	China	127 (62/65)	Both genders Median: 57/57	IV	HCC (Stage 0-B ^b^)	Solid	^131^I-metuximab	CD147/HAb18G
Quackenbush et al. (2015) [25]	USA	14 (8/6)	Both genders Mean: 53.1/58.2	III	FL	Non-solid	^131^I-tositumomab	CD20
Zhao et al. (2016) [26]	China	96 (47/49)	Both genders Mean: 57	N/A	NSCLC (Stage II and IIIa)	Solid	^131^I-chTNT	DNA/Histone H1
Li et al. (2020) [27]	China	156 (78/78)	Both genders Median: 53/53	II	HCC	Solid	^131^I-metuximab	CD147/HAb18G
López-Guillermo et al. (2022) [28]	Spain	126 (64/62)	Both genders Median: 52/53	II	FL (Stages II, III, or IV)	Non-solid	^90^Y-ibritumomab tiuxetan	CD20
Tagawa et al. (2023) [29]	USA	55 (38/17)	Male Median: 68	II	Prostate cancer	Solid	^177^Lu-J591	PSMA
Gyurkocza et al. (2024) [30]	USA	153 (76/77)	Both genders Median: 64/66	III	AML	Non-solid	^131^I-apamistamab	CD45
Ladetto et al. (2024) [31]	Italy	141 (71/70)	Both genders Median: 56/58	III	FL (Stage III/IV)	Non-solid	^90^Y-ibritumomab tiuxetan	CD20
Laoruangroj et al. (2024) [32]	USA	20 (10/10)	Both genders Median: 59/61	III	FL (Stages I, II, III, IV)	Non-solid	^90^Y-ibritumomab tiuxetan	CD20

I: intervention; C: comparator; NHL: non-Hodgkin’s lymphoma; DLBCL: diffuse large B-cell lymphoma; EOC: epithelial ovarian cancer; HCC: hepatocellular carcinoma; FL: follicular lymphoma; NSCLC: non-small-cell lung cancer; AML: acute myeloid leukaemia; ^a^: according to International Federation of Gynecology and Obstetrics (FIGO); MUC1: mucin 1; EGFR: epidermal growth factor receptor; CEA: carcinoembryonic antigen; PSMA: prostate-specific membrane antigen. ^b^: according to Barcelona Clinic Liver Cancer staging system (BCLC) classifications.

**Table 2 cancers-17-01413-t002:** Treatment outcomes and key findings from the included clinical trials.

Clinical Trial/Study (Year)	Key Treatment Outcomes [Intervention (I) Versus Control (C)]	Key Findings	RIT Treatment with Other Therapy
Witzig et al. (2002) [13]	Median TTP(ITT):11.2 vs. 10.1 months (*p* = 0.173).	^90^Y-ibritumomab tiuxetan was well tolerated and significantly improved ORR and CR vs. rituximab alone.	Rituximab
Gibson, A. D. (2002) [33]	Median TTP (ITT): 10.6 vs. 10.1 months (*p* = 0.425)	^90^Y-ibritumomab tiuxetan is effective for R/R low-grade or follicular NHL vs. rituximab.	Rituximab
Gordon et al. (2004) [34]	Median TTP (ITT): 10.6 vs. 10.1 months (*p* = 0.41)	RIT with ^90^Y-ibritumomab tiuxetan is effective in follicular NHL, especially for patients achieving CR.	Rituximab
Davis et al. (2004) [14]	Median PFS: 6.3 vs. 5.5 months (*p* = 0.016)	Combination of ^131^I and tositumomab significantly improved OR, CR, and TTP in relapsed NHL.	None
Verheijen et al. (2006) [15]	RR of death (ITT): 1.159 (*p* = 0.4033); 31.3% vs. 27.4% deaths	A single intrapleural dose of ^90^Y-muHMFG1 did not prolong survival or time to relapse in EOC patients.	Chemotherapy
Oei et al. (2007) [35]	Intraperitoneal relapse-free survival (HR = 0.31; *p* = 0.002)	No survival benefit for intraperitoneal RIT as consolidation treatment in EOC.	Chemotherapy
Wygoda et al. (2006) [16]	Median OS: 14 months; no significant OS/DFS difference (*p* = 0.23)	Concomitant radiotherapy and RIT (with anti-EGFR ^125^I-Mab 425) was superior to radiotherapy alone in high-grade gliomas.	Teleradiotherapy
Xu et al. (2007) [17]	Median follow-up:12.3 (range = 2 to 13, mean = 10.99) months. 3-month OS: 100% vs. 93.1%; 6-month OS: 96.7% vs. 75.9%; 9-month OS: 90% vs. 69.0% (I); 12-month OS: 82.5% vs. 61.9% (*p* = 0.0289).	^131^I-metuximab after liver transplantation was well tolerated, reduced recurrence, and prolonged survival in HCC.	OLT
Morschhauser et al. (2008) [18]	Median PFS: 36.5 vs. 13.3 months (HR = 0.465; *p* < 0.0001)	Consolidation with ^90^Y-ibritumomab tiuxetan after first-line therapy significantly prolonged 2-year PFS.	Rituximab
Morschhauser et al. (2013) [36]	Median PFS: 4.1 vs. 1.1 years (HR = 0.47; *p* < 0.001	First-line consolidation therapy with ^90^Y-ibritumomab is valuable for patients with advanced FL, providing a durable PFS benefit.	Rituximab
Goff et al. (2009) [19]	Median PFS: 3 years vs. 13 months (*p* < 0.0001; HR = 0.465)	^90^Y-ibritumomab consolidation deepened molecular response and prolonged PFS.	Rituximab
Sultana et al. (2009) [20]	Median OS: 5.2 months; OS difference not significant (*p* = 0.79)	^131^I-Kab201 monotherapy was comparable to gemcitabine; combination therapy might improve survival.	N/A
Shimoni et al. (2012) [21]	2-yr PFS: 48%; 2-yr OS: 77% overall; 91% vs. 62% (*p* = 0.05)	Adding ^90^Y-ibritumomab tiuxetan to BEAM chemotherapy may improve outcomes in ASCT conditioning.	BEAM, rituximab, ASCT
Press et al. (2013) [22]	2-/5-yr PFS: 80%/66% vs. 76%/60% (*p* = 0.11)	No clear PFS benefit of CHOP-RIT over R-CHOP; both arms had excellent PFS/OS.	CHOP
Shadman et al. (2018) [37]	10-yr PFS: 56% vs. 42% (*p* = 0.011); 10-yr OS: 75% vs. 81% (*p* = 0.12)	CHOP-RIT extended PFS but did not improve OS vs. CHOP-R alone.	CHOP
Vose et al. (2013) [23]	2-yr PFS: ~48% vs. ~48% (*p* = 0.94); 2-yr OS: ~66% vs. ~61% (*p* = 0.38)	Adding RIT to AHCT did not show additional benefit in relapsed DLBCL.	BEAM
Bian et al. (2014) [24]	1-/2-yr OS: 93.5%/84.7% vs. 90.1%/76.4% (HR = 0.66; *p* = 0.30)	^131^I-metuximab post-RFA reduces recurrence in HCC; CD147-targeted strategy shows promise.	RFA
Quackenbush et al. (2015) [25]	Median PFS: NR vs. 9 months (*p* = 0.0705); OS: all vs. 3/6 alive (*p* = 0.0272)	^131^I-tositumomab provided durable clinical benefit in relapsed FL with selected patients.	Tositumomab
Zhao et al. (2016) [26]	1-/2-yr OS: ~83%/53% vs. ~80%/49% (*p* > 0.05); median survival: 29.1 vs. 23.0 months (*p* < 0.05)	^131^I-chTNT RIT + PMCT improved survival in NSCLC, with efficacy comparable to adjuvant chemotherapy.	PMCT, follow-up chemotherapy
Li et al. (2020) [27]	5-yr RFS: 43.4% vs. 21.7% (*p* < 0.0001); 5-yr OS: 61.3% vs. 35.9% (*p* < 0.0001)	^131^I-metuximab adjuvant therapy significantly improved RFS/OS in CD147+ HCC after hepatectomy.	Hepatectomy
López-Guillermo et al. (2022) [28]	10-yr PFS: 50% vs. 56% (*p* = 0.19); OS: 78% vs. 84.5%	In FL responding to R-CHOP, RIT did not significantly differ in PFS/OS, with potential late toxicities.	R-CHOP
Tagawa et al. (2023) [29]	Median MFS: 23.8 vs. 20.8 months; 18-mo MFS: 50% vs. 24% (*p* = 0.066)	^177^Lu-J591 (anti-PSMA) showed longer MFS vs. ketone/HC in prostate cancer, though best radionuclide remains unclear.	Ketoconazole
Gyurkocza et al. (2024) [30]	Median OS: ~6.3 vs. ~5.9 months (*p* = 0.59); Median EFS: 3.2 vs. 0 month (*p* < 0.0001)	^131^I-apamistamab showed higher durable CR in elderly R/R AML vs. standard care, addressing an unmet need.	alloHCT
Ladetto et al. (2024) [31]	Median PFS: 78 vs. 62 months (HR = 1.11, *p* = 0.6662); OS not reached in both arms	ASCT offered no advantage over RIT; RIT consolidation yields excellent disease control with less toxicity.	Rituximab, immunochemotherapy
Laoruangroj et al. (2024) [32]	Median PFS: ~29.9 months vs. NR (*p* = 0.431)	Both rituximab alone and rituximab + single-dose RIT were highly effective in asymptomatic FL.	Rituximab

Note: Table 2 displays the treatment outcomes and the most important results of the 25 clinical trials that were initially included. OS: overall survival; PSF: progression-free survival; DFS: disease-free survival; EFS: event-free survival; TTP: time to progression; RR: relative risk; RFS: recurrence-free survival; MFS: metastasis-free survival; I: intervention; C: comparator; ITT: intention to treat; CI: confidence interval; HR: hazard ratio; NR: not reached; NE: not estimated; OR: overall response; ORR: overall response rate; CR: complete response; BEAM: carmustine, etoposide, cytarabine, and melphalan; ASCT: autologous stem cell transplantation; AHCT: autologous hematopoietic cell transplantation; CHOP: cyclophosphamide, doxorubicin, vincristine, and prednisone; R-CHOP: rituximab, cyclophosphamide, doxorubicin, vincristine, and prednisone; R/R: relapse and refractory; RFA: radiofrequency ablation; PMCT: percutaneous microwave coagulation therapy; *p* ≤ 0.05: statistically significant; *p* ≥ 0.05: not statistically significant; ASCT: autologous stem cell transplantation; RFA: radiofrequency ablation; alloHCT: allogeneic hematopoietic cell transplantation; OLT: orthotopic liver transplantation.

**Table 3 cancers-17-01413-t003:** Comparison and combination of RIT therapy with other therapies.

Clinical Trial/Study (Year)	Comparator Treatment	RIT Treatment with Other Therapy	Outcome
Witzig et al. (2002) [13]	Rituximab alone	Rituximab	RIT combination therapy was superior to rituximab alone
Davis et al. (2004) [14]	Unlabeled tositumomab	None	Radiolabelled tositumomab improved treatment outcome
Verheijen et al. (2006) [15]	Chemotherapy alone	Chemotherapy	RIT combination therapy was similar to chemotherapy alone
Wygoda et al. (2006) [16]	Teleradiotherapy alone	Teleradiotherapy	RIT combination therapy was not superior to teleradiotherapy alone
Xu et al. (2007) [17]	Placebo (physiological saline)	OLT	RIT combination therapy was superior to placebo
Morschhauser et al. (2008) [18]	No consolidation treatment ^a^	Rituximab	RIT combination therapy was superior to no consolidation therapy
Goff et al. (2009) [19]	No consolidation treatment ^a^	Rituximab	RIT combination therapy was superior to no consolidation therapy
Sultana et al. (2009) [20]	N/A (compare different routes of RIT administration)	N/A	N/A
Shimoni et al. (2012) [21]	BEAM alone	BEAM, rituximab, ASCT	RIT combination therapy was superior to BEAM alone
Press et al. (2013) [22]	R-CHOP	CHOP	RIT combination therapy was similar to R-CHOP
Vose et al. (2013) [23]	R-BEAM	BEAM	RIT combination therapy was not superior to R-BEAM
Bian et al. (2014) [24]	RFA alone	RFA	RIT combination therapy was superior to RFA alone
Quackenbush et al. (2015) [25]	Rituximab alone	Tositumomab	RIT combination therapy was superior to rituximab alone
Zhao et al. (2016) [26]	Surgery, chemotherapy, radiotherapy	PMCT, follow-up chemotherapy	RIT combination therapy was superior to surgery, chemotherapy, and radiotherapy
Li et al. (2020) [27]	No adjuvant treatment ^a^	Hepatectomy	RIT combination therapy was superior to no adjuvant treatment
López-Guillermo et al. (2022) [28]	Rituximab alone	R-CHOP	No significant difference between RIT combination therapy and rituximab alone
Tagawa et al. (2023) [29]	^111^In-J591, ketoconazole	Ketoconazole	RIT combination therapy prolonged survival but with more toxicity
Gyurkocza et al. (2024) [30]	Salvage therapy followed by standard-of-care alloHCT	alloHCT	RIT combination therapy was superior to standard-of-care alloHCT
Ladetto et al. (2024) [31]	ASCT	Rituximab, immunochemotherapy	RIT combination therapy was superior to ASCT
Laoruangroj et al. (2024) [32]	Rituximab alone	Rituximab	No significant difference between RIT combination therapy and rituximab alone

Note: Table 3 displays the final 20 clinical trials that passed the quality assessment and were included in the systematic analysis. CHOP: cyclophosphamide, doxorubicin, vincristine, and prednisone; R-CHOP: rituximab, cyclophosphamide, doxorubicin, vincristine, and prednisone; BEAM: carmustine, etoposide, cytarabine, and melphalan; R-BEAM: rituximab, carmustine, etoposide, cytarabine, and melphalan; ASCT: autologous stem cell transplantation; RFA: radiofrequency ablation; alloHCT: allogeneic hematopoietic cell transplantation; OLT: orthotopic liver transplantation; PMCT: percutaneous microwave coagulation therapy; ^a^: trials in which RIT was used as a consolidation therapy or as an adjuvant therapy after first-line therapy.

**Table 4 cancers-17-01413-t004:** Adverse events reported in clinical trials included.

Clinical Trial/Study (Year)	Nonhaematological AEs	Haematological AEs
Witzig et al. (2002) [13] ^a^	Grades 1/2 (Intervention/Control): - Cough: 15%/7%; - Dyspnea: 15%/7%; - Nausea: 43%/19% - Vomiting: 19%/7%; - Anorexia: 11%/3%	Grade 3/4 (Intervention only): - ANC: 57%; - Platelets: 60% MDS: one case
Davis et al. (2004) [14] ^a^	(Intervention/Control): - Nausea: 48%/17%; – Rash: 31%/14%; - Chills: 24%/19% - Pain: 21%/28% (also overall drug-related AEs: 100% vs. 89% [all grades], 71% vs. 31% [grade 3/4], serious: 33% vs. 14%)	Grade 3/4 (Intervention/Control): - ANC: 33%/8%; - Platelets: 33%/0% MDS/AML (Intervention): three cases (5%)
Verheijen et al. (2006) [15] ^a^	(Intervention/Control): - Nausea: 40%/19%; - Fatigue: 34%/21%; - Arthralgia: 31%/19% - Myalgia: 25%/7%; - Abdominal pain: 25%/16% - Rash: 17%/5%; – Diarrhoea: 17%/8%; - Vomiting: 17%/8%	Grade 3/4: - Thrombocytopenia (Intervention only): 24.3%
Wygoda et al. (2006) [16]	Not Reported	Not Reported
Xu et al. (2007) [17]	Not Reported	Not Reported
Morschhauser et al. (2008) [18] ^a^	Grade 1/2 (Intervention only): - Fatigue: 32.8%; - Nasopharyngitis: 19.1% - Nausea: 18.1%; - Asthenia: 14.2%; - Arthralgia: 11.8%; - Cough: 11.3%; - Headache: 11.3%; - Diarrhoea: 10.8%; - Pyrexia: 10.3% Grade 3/4 (Intervention/Control): - Infections: 7.9%/2.4%; - Pyrexia: 3%/0% - Hypertension: 2.9%/0.5%	Grade 3/4 (Intervention/Control): - Lymphopenia (60.3%/10.8%); - Neutropenia (66.7%/2.5%); - Thrombocytopenia (60.8%/0%); - Anaemia (3.4%/0%). AML (Intervention): one case
Goff et al. (2009) [19]	N/A	N/A
Sultana et al. (2009) [20] ^a^	Not reported	Grade 3/4 drug-related: - Lymphopenia (*n* = 5); - Thrombocytopenia (*n* = 6) - Leukopenia (*n* = 4); - Neutropenia (*n* = 3)
Shimoni et al. (2012) [21]	Organ toxicity (≥Grade 3): - Mucositis: 15 (I) vs. 9 (C) - Pneumonia/fungal infection: 6 (I) vs. 1 (C)	Not reported
Press et al. (2013) [22]	Cardiovascular events (Grade 3–5): 3% (I) vs. 7% (C)	(Intervention/Control): - Thrombocytopenia: 18% vs. 2% - Febrile neutropenia: 10% vs. 16% - AML/MDS: 3% vs. 1%
Vose et al. (2013) [23] ^b^	Any Grade 3–5 nonhaematological toxicity: 65% (I) vs. 43% (C) - Mucositis: 52% (I) vs. 18% (C)	MDS/AML: two cases (I) vs. one case (C)
Bian et al. (2014) [24] ^b^	(Intervention/Control): - Pleural effusion: 20.0% vs. 9.4% - Increased ALT: 28.3% vs. 42.2% - Increased AST: 26.7% vs. 43.8%	(Intervention/Control): - Decreased WBC: 66.7% vs. 50.0% - Grade 3 decreased platelet count: 18.3% vs. 7.8%
Quackenbush et al. (2015) [25] ^b^	Intervention: headache, nausea, vomiting, cough Control: pyrexia, infusion-related reactions	Intervention: - Neutropenia (*n* = 1), thrombocytopenia (*n* = 3), leukopenia (*n* = 4), lymphopenia (*n* = 4) Control: none reported *(Serious AEs: four cases per arm)*
Zhao et al. (2016) [26]	Radioactive esophagitis: 4.25% (I) vs. 20.4% (C)	Myelotoxicity: 14.89% (I) vs. 8.16% (C)
Li et al. (2020) [27] ^b^	- Fever: 3%; - Fatigue: 3%; - Nausea/vomiting: 1% - Increased bilirubin/ALT/AST/ALP: 1%	Decreased WBC/platelet counts: 2% (Intervention)
López-Guillermo et al. (2022) [28]	(Intervention/Control): - Infectious complications: 2% vs. 13% - Second neoplasms: 14% vs. 3%	(Intervention/Control): - Neutropenia: 9% vs. 3% - Thrombocytopenia: 8% vs. 0% (*p* = 0.055) - MDS/AML/HL: 9% vs. 0% (*p* = 0.028)
Tagawa et al. (2023) [29]	(Intervention/Control): - Abdominal pain: 0% vs. 11%; - Increased ALT: 3.3% vs. 2% - Diarrhoea: 0% vs. 22%; - All-cause mortality: 34.21% vs. 29.41%; - Serious AEs: 5.26% vs. 0%	(Intervention/Control): - Neutropenia: 57% vs. 11% - Thrombocytopenia: 77% vs. 11%
Gyurkocza et al. (2024) [30] ^b^	Not separately detailed	Febrile neutropenia: 18.1% (I) vs. 22.4% (C) - Treatment-related deaths: 4.2% (I) vs. 5.3% (C) *(Serious AEs: 30.6% vs. 31.6%)*
Ladetto et al. (2024) [31] ^b^	**Consolidation:** - Nonhaematological AEs: 5% (I) vs. 37% (C) (mostly GI disorders) **Maintenance:** N/A	**Consolidation:** - Haematological AEs: 46% (I) vs. 93% (C) (*p* < 0.0001) **Maintenance:** - Haematological AEs: 13% (I) vs. 25% (C) (*p* = 0.168) - Secondary MDS/AML: two cases (I) vs. three cases (C)
Laoruangroj et al. (2024) [32] ^b^	Not reported	Grade 3/4 haematological AEs: - Leukopenia: 40% (I) vs. 0% (C); - Neutropenia: 40% (I) vs. 0% (C) - Thrombocytopenia: 40% (I) vs. 0% (C) - Anaemia: 10% (I) vs. 0% (C)

I: intervention; C: comparator; ANC: absolute neutrophil count; MDS: myelodysplastic syndrome; AML: acute myeloid leukaemia; WBC: white blood cell; HL: Hodgkin’s lymphoma; ALT: alanine aminotransferase; AST: aspartate aminotransferase; ALP: alkaline phosphatase. ^a^: use CTC version 2.0. ^b^: use CTCAE versions 3.0 or 4.0.

## Data Availability

Data will be available upon reasonable request to the corresponding author.

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
