# Peer review of "Assessing the Clinical Effectiveness of Radioimmunotherapy with Combined Radionuclide/Monoclonal Antibody Conjugates in Cancer Treatment: Insights from Randomised Clinical Trials"

_cancers, 2025, doi:10.3390/cancers17091413_

Round 1
Reviewer 1 Report
Comments and Suggestions for Authors
Overall, the manuscript is a well-conducted systematic review and meta-analysis that highlights the efficacy and safety of RIT in cancer treatment. However, by addressing the following limitations, the review could offer a more comprehensive and clinically relevant evaluation of RIT.
-The review focuses on survival-related outcomes (e.g., OS, PFS) but does not include other critical endpoints such as tumor response rates, quality of life, or economic outcomes. Including these metrics would provide a more comprehensive assessment of RIT’s impact.
-While the study reports common adverse events (e.g., hematological toxicities, nausea, fatigue), it does not provide detailed recommendations on how to manage these toxicities in clinical practice. A more in-depth discussion on toxicity mitigation strategies would enhance the clinical relevance of the review.
-The study highlights the dominance of CD20-targeted RIT in NHL but does not sufficiently explore the potential of other tumor-associated antigens, especially for solid tumors. Expanding the discussion on novel antigen targets could provide a more forward-looking perspective.
Author Response
Response to the reviewer 1 comments
Point 1: Overall, the manuscript is a well-conducted systematic review and meta-analysis that highlights the efficacy and safety of RIT in cancer treatment. However, by addressing the following limitations, the review could offer a more comprehensive and clinically relevant evaluation of RIT.
Author’s response 1: Thank you for your valuable feedback and suggestions. The manuscript has been revised to address the reviewer’s comments.
Point 2: -The review focuses on survival-related outcomes (e.g., OS, PFS) but does not include other critical endpoints such as tumor response rates, quality of life, or economic outcomes. Including these metrics would provide a more comprehensive assessment of RIT’s impact.
Author’s response 2: We appreciate the reviewer’s emphasis on additional outcomes such as tumour response, quality of life, and economic outcomes. As mentioned in the limitations section, the primary objective of the study was to evaluate the efficacy of RIT in terms of clinical outcomes (e.g., OS, PFS). While we recognise the importance of comprehensive endpoints, including quality of life and economic outcomes, we intentionally confined our scope to prevent the study from becoming overly broad, which could dilute the depth of our analysis on survival outcomes. Nonetheless, these additional measures are indeed valuable and we agree that they merit separate, dedicated investigations—such as a full health economic evaluation or studies focused on patient-reported outcomes.
Point 3: -While the study reports common adverse events (e.g., hematological toxicities, nausea, fatigue), it does not provide detailed recommendations on how to manage these toxicities in clinical practice. A more in-depth discussion on toxicity mitigation strategies would enhance the clinical relevance of the review.
Author’s response 3: Thank you for your valuable feedback on providing more detailed management strategies for common adverse events. Regarding the common adverse events (AEs), these AEs are inherent in the treatment process due to the intrinsic toxicity of the therapy (i.e., radionuclides). As already mentioned in the paper, it is important to treat patients with caution, especially in the case of fragile patients (e.g., patients with low blood counts, elderly patients). We add the close monitoring of patients’ conditions and the provision of certain medications when it is necessary. In addition to these management methods, we felt that the discovery of safer radionuclides and the development of novel RIT without compromising efficacy was a fundamental part of addressing this issue.
To further enhance clinical relevance, we have included additional recommendations for managing toxicities in practice. These strategies include: routine complete blood count (CBC) monitoring and timely dose adjustments for hematological toxicities; prophylactic and/or rescue use of growth factors (e.g., G-CSF) to mitigate neutropenia; administration of antiemetics to prevent or control nausea and vomiting; proactive fluid and electrolyte management in patients at risk of dehydration or renal complications; careful assessment of the need for blood product transfusions in cases of severe anemia or thrombocytopenia; and close collaboration among oncologists, pharmacists, and supportive care teams to ensure individualised management for older adults or those with comorbidities. These steps, coupled with ongoing research into safer radionuclides and novel RIT approaches, are essential for balancing treatment efficacy with optimal patient safety and comfort. Please see page 39-40.
Point 4: -The study highlights the dominance of CD20-targeted RIT in NHL but does not sufficiently explore the potential of other tumor-associated antigens, especially for solid tumors. Expanding the discussion on novel antigen targets could provide a more forward-looking perspective.
Author’s response 4: Thank you for your insightful comment. We acknowledge that the review places substantial emphasis on CD20-targeted RIT in NHL due to the abundance of robust clinical data. However, we agree that the potential of other tumor-associated antigens—particularly for solid tumors—warrants further exploration to provide a forward-looking perspective.
In the revised manuscript, we have expanded the discussion on several important antigens beyond CD20:
We have added at the end of the second paragraph, “The antigen targeted by 131I-metuximab, CD147, showed positive results in three liver cancer trials, demonstrating that CD147-targeted RIT was effective in treating liver cancer; further research in liver cancer may continue to focus on the CD147 antigen. However, other tumor-associated antigens for solid tumors did not show encouraging results. Mucin 1 (MUC1) was a target for the treatment of epithelial ovarian tumor in the included trial and for pancreatic tumor in another trial [43], but this trial was terminated prematurely without a positive outcome for RIT. Epithelial growth factor receptor (EGFR) was a target for the treatment of gliomas in the included trial, while Wygoda et al [16] concluded that EGFR-targeted RIT was not effective in high-grade gliomas, another phase II trial showed encouraging results in glioblastoma multiforme [44]. However, a large RCT study is still needed to confirm its effectiveness in the treatment of brain tumors. Prostate-specific membrane antigen (PSMA) has been a main target for treating prostate cancer, it has shown encouraging results when used as radioligand therapy (i.e., 177Lu-PSMA-617 [45], however, Tagawa et al [29] concluded that high toxicity followed treatment when used as RIT even though it prolonged survival. This highlights the need to further improve the safety profile of PSMA-targeted RIT for translation to the clinic.” Please see page 34.
Reviewer 2 Report
Comments and Suggestions for Authors
This paper reports the meta-analysis result of the assessment the effectiveness of RIT for various cancers by analysis the randomized controlled clinical studies over the last two decades. The study provides useful information on the most common cancer types treated by RIT and potential directions for optimizing cancer treatment based on RIT. Overall, the study is interesting and the conclusion is solid. The manuscript can be accepted for publication after addressing the following minor concern.
- It would be helpful that authors also discuss the clinical treatment impact of different radionuclides used for RIT.
Author Response
Response to the reviewer 2 comments
Point 1: This paper reports the meta-analysis result of the assessment the effectiveness of RIT for various cancers by analysis the randomized controlled clinical studies over the last two decades. The study provides useful information on the most common cancer types treated by RIT and potential directions for optimizing cancer treatment based on RIT. Overall, the study is interesting and the conclusion is solid. The manuscript can be accepted for publication after addressing the following minor concern.
Author’s response 1: Thank you. We have revised to each concern in a point-by-point basis.
Point 2: It would be helpful that authors also discuss the clinical treatment impact of different radionuclides used for RIT.
Author’s response 2: Thank you for your insightful comment. However, since the clinical treatment impact of RIT is related not only to the radionuclide used, but also to the monoclonal antibody used, and the combination between the two, it is difficult to directly evaluate the efficacy by itself. Instead, we discussed the commonly used radionuclides and other novel radionuclides used, and their characteristics that may affect treatment effects.
We have added a section discussing the effects of different radionuclides.
We have added a paragraph after the second paragraph, “In addition to the antigen, the radionuclides used were also important in determining the efficacy of RIT. Most of the included trials used yttrium-90 (Y-90) and iodine-131 (I-130), with two trials using iodine-125 (I-125) and lutetium-177 (Lu-177) respectively. Importantly, all of the included trials using Y-90 were for the treatment of lymphoma, whereas the trials using I-130 were also for the treatment of solid tumours, mainly liver cancer, as well as lymphoma. I-125 and Lu-177 were both used exclusively for the treatment of solid tumours in the present study. Y-90, I-130 and Lu-177 all have a relatively short half-life of about 2-8 days, while I-125 has a longer half-life of about 59.4 days. Interestingly, it is said that the properties of I-125 make it a preferred isotope for radiotherapy in the treatment of brain tumours and prostate cancer, but the prostate cancer treated in this study used Lu-177 instead [46]. Furthermore, contrary to the suggestion by Leonard et al [47] that Y-90 would be preferable to 131-I for patients with solid tumors, 131-I has been used extensively for the treatment of liver cancer, with encouraging results in all cases. This highlights that 131-I would be useful for the treatment of solid tumors. In addition, the more encouraging results for lymphoma in the studies using Y-90 rather than 131I suggest that Y-90 may be more suitable for the treatment of non-solid tumours. In general, the choice of a suitable radionuclide is important for the clinical treatment effect of RIT, but not only the chemical properties themselves, but also the availability, cost-effectiveness and safety profile should be considered.” See pages 35-36.
Reviewer 3 Report
Comments and Suggestions for Authors
In the manuscript, Chen and coauthors conducted a comprehensive meta-analysis evaluating the clinical effectiveness of radioimmunotherapy (RIT) in treating various blood and solid cancers. The authors analyzed data from 20 published clinical trials and found that while RIT improves treatment outcomes in blood cancers when combined with other established or experimental therapies, it is less effective in solid tumors.
Overall, the manuscript is well-written, easy to read, and maintains a neutral tone without obvious bias. The discussion appropriately summarizes the findings, presents both strengths and limitations, and describes their implications. However, the manuscript lacks significant novelty and does not provide substantial new insights into the efficacy and clinical application of RIT.
Several minor issues have been identified in the manuscript. The following comments may help improve its clarity and readability:
- Please provide full definitions for abbreviations when they first appear, such as RIT in line 19 and NHL in line 49.
- There are some punctuation errors, including a comma between “60%” and “(n=12)” in line 197, a comma after “cancer” in line 204, and a comma between “Comparator” and “NHL” in line 215.
- Please define “P” and “p” in Table 2. What do these values represent? Statistical significance?
- It is recommended to add a column in Table 3 to compare the outcomes of combination treatments with those of comparator treatments, which would make it easier for readers to identify the key information.
- The image quality of Figures 2.1 and 2.2 is very low; please provide higher-resolution versions.
- How are non-hematologic AEs and hematologic AEs defined and classified? Why are certain AEs, such as nausea, pain, and cough, included in both categories?
- Please remove “Supplementary Table 3” in line 313.
Author Response
Response to the reviewer 3 comments
Point 1: In the manuscript, Chen and coauthors conducted a comprehensive meta-analysis evaluating the clinical effectiveness of radioimmunotherapy (RIT) in treating various blood and solid cancers. The authors analyzed data from 20 published clinical trials and found that while RIT improves treatment outcomes in blood cancers when combined with other established or experimental therapies, it is less effective in solid tumors.
Overall, the manuscript is well-written, easy to read, and maintains a neutral tone without obvious bias. The discussion appropriately summarizes the findings, presents both strengths and limitations, and describes their implications. However, the manuscript lacks significant novelty and does not provide substantial new insights into the efficacy and clinical application of RIT. Several minor issues have been identified in the manuscript. The following comments may help improve its clarity and readability:
Authors’ response: Thank you very much for your insightful and constructive comments. The manuscript has been revised to address the reviewers’ comments. We acknowledge your concern regarding the manuscript's novelty and agree that, at present, our meta-analysis does not substantially advance novel insights into RIT efficacy, particularly for solid tumors. This primarily reflects the current landscape of RIT research, which remains predominantly focused on blood cancers, especially lymphomas, where results are consistent across multiple trials. Indeed, there remains limited data regarding the efficacy of RIT in solid tumors, highlighting a critical gap in the existing literature. We agree that future research should prioritise evaluating RIT as a stand-alone therapy to generate clearer insights into its independent therapeutic potential.
We have carefully addressed the minor issues you pointed out, improving clarity and readability as suggested.
Point 2: Please provide full definitions for abbreviations when they first appear, such as RIT in line 19 and NHL in line 49.
Authors’ response 2: We have clarified the abbreviations used in the manuscript. The full form of RIT ("radioimmunotherapy") is provided on page 2, and NHL ("Non-Hodgkin lymphoma") has been introduced on page 3.
Point 3: There are some punctuation errors, including a comma between “60%” and “(n=12)” in line 197, a comma after “cancer” in line 204, and a comma between “Comparator” and “NHL” in line 215.
Author’s response 3: Thank you. Please see page 11 of the “Characteristics of included clinical trials” section, where the commas have been removed.
The revised sentence now reads:
“Of the clinical trials included in the review, approximately 60% (n = 12) were for non-solid tumours”, “Eight (40%) studies were on solid tumours, of which 4 (20%) were for lung cancer”.
Please see page 15 of the bottom of Table 1 where the comma between “Comparator” and “NHL” has been replaced by “;”.
Point 4: Please define “P” and “p” in Table 2. What do these values represent? Statistical significance?
Author’s response 4: We apologise for the confusion. The term ‘p’ refers to the “probability value (p-value)”, which indicates the level of statistical significance. The manuscript text and Table 2 have been updated accordingly. Please see the revised manuscript (page 19) and Table 2 for details.
Point 5: It is recommended to add a column in Table 3 to compare the outcomes of combination treatments with those of comparator treatments, which would make it easier for readers to identify the key information.
Authors’ response 5: Thank you for the suggestion. We have added a column referring to the clinical “outcomes” of combined treatments as comparative treatment strategies, and these revised results are now reflected in the updated manuscript.
The revised texts now read as
“Table 3 summarises findings from various clinical trials evaluating the effectiveness of radioimmunotherapy (RIT) in combination with other treatments such as chemotherapy, immunotherapy, surgery, and targeted therapy across multiple cancer types. Notably, RIT combination therapy consistently demonstrated superiority over several comparator treatments, including rituximab alone, placebo, standard chemotherapy regimens (e.g., BEAM, R-BEAM), radiofrequency ablation (RFA), salvage therapies, autologous stem cell transplantation (ASCT), and multimodal approaches involving surgery, chemotherapy, and radiotherapy (Witzig et al., 2002; Xu et al., 2007; Shimoni et al., 2012; Bain et al., 2014; Zhao et al., 2016; Gyurkocza et al., 2024; Ladetto et al., 2024). Conversely, outcomes from RIT combinations were comparable or showed no significant difference compared to standard chemotherapy alone, R-CHOP, or rituximab monotherapy in other trials (Vriehling et al., 2005; Press et al., 2013; Vose et al., 2013; López-Guillermo et al., 2022; Laoruangroj et al., 2024). However, caution is warranted, particularly in solid tumors, as the data from Tagawa et al. (2023) indicated prolonged survival with RIT combinations but at the expense of increased toxicity. Thus, while RIT combination therapies appear promising and consistently beneficial for hematologic malignancies, their utility in solid tumors remains uncertain, highlighting the need for further investigation into targeted antigen selection and patient stratification to optimise clinical benefits.” Please see pages 21-22.
Point 6: The image quality of Figures 2.1 and 2.2 is very low; please provide higher-resolution versions.
Author’s response 6: Thank you. As suggested, we have how added the high quality of images.
Point 7: How are non-hematologic AEs and hematologic AEs defined and classified? Why are certain AEs, such as nausea, pain, and cough, included in both categories?
Author’s response 7: We have added the criteria for defining grade 3/4 hematologic AEs and revised the corresponding table, please see pages 26 and 30. We have removed nonhematologic AEs from the haematologic AEs category and replaced it with the correct AEs, please see page 28.
Point 8: Please remove “Supplementary Table 3” in line 313.
Author’s response 8: We apologise for any confusion caused. We assessed the quality of the selected papers using the Joanna Briggs Institute (JBI) Critical Appraisal Tool for RCTs, while our study aimed to select papers based on the clinical effectiveness of RIT through randomised controlled trials that align with our study objectives. These quality assessment results are presented in Supplementary Table 3, which is critically important for the relevant audience to include the supplementary materials.
Reviewer 4 Report
Comments and Suggestions for Authors
The major purpose of the manuscript presented by Chen et al. was to systematically review the clinical effects and safety profiles of radioimmunotherapy focusing on combined radionuclide and monoclonal antibody conjugates mainly targeting the CD20 antigen. In more detail, on a total 20 randomized controlled trials, meta-analysis of progression-related outcomes yielded a pooled hazard ratio (HR) of 0.48 (95% CI: 53 0.39–0.59), indicating a 52% reduction in the risk of progression with most studies covering non-Hodgkin’s lymphoma. Overall survival outcomes were more variable, with a pooled OS HR of 0.80 (95% CI: 0.60–1.07). In addition, hematological and nonhematological toxicities were common, but reversible.
The topic the review, in principle, is highly interesting and of clinical relevance, but warrants further investigation especially for solid tumors. There are, however, a number of shortcomings as listed successive that limit the impact of the manuscript.
Major points of criticism:
- Title: Authors mainly focused on radionuclide/monoclonal antibody conjugate therapy. This should clearly be given in the title, for instance by “Assessing the Clinical Effectiveness of Radioimmunotherapy with combined Radionuclide/Monoclonal Antibody Conjugates in Cancer Treatment: Insights from Randomised Clinical Trials”
- Simple summary an abstract section: Please harmonize number of patients 3562 or 3000?
- The comparative analysis in Table 3 indicates that radioimmunotherapy is predominantly integrated as part of combination regimens rather than given as mono therapy. Accordingly, treatment regimens given in table 3 should be included in table 2 to increase readability.
- Authors stressed the point that the monoclonal antibodies target tumor-associated antigens or those expressed by the tumor microenvironment, but failed to cover this information in the tables, e.g. table 1.
- It is not tractable to the reviewer, why authors indicate a number 20 studies to be included in the systematic analyses (figure 1), but displayed a total of 25 trials in table 2 and total of 20 studies in table 3. This discrepancy has to be solved.
Author Response
Response to the reviewer 4 comments
Point 1: The major purpose of the manuscript presented by Chen et al. was to systematically review the clinical effects and safety profiles of radioimmunotherapy focusing on combined radionuclide and monoclonal antibody conjugates mainly targeting the CD20 antigen. In more detail, on a total 20 randomized controlled trials, meta-analysis of progression-related outcomes yielded a pooled hazard ratio (HR) of 0.48 (95% CI: 53 0.39–0.59), indicating a 52% reduction in the risk of progression with most studies covering non-Hodgkin’s lymphoma. Overall survival outcomes were more variable, with a pooled OS HR of 0.80 (95% CI: 0.60–1.07). In addition, hematological and nonhematological toxicities were common, but reversible.
The topic the review, in principle, is highly interesting and of clinical relevance, but warrants further investigation especially for solid tumors. There are, however, a number of shortcomings as listed successive that limit the impact of the manuscript.
Author’s response 1: Thank you for your comments. We have responded to each concern in a point-by-point manner.
Major points of criticism:
Point 2: Title: Authors mainly focused on radionuclide/monoclonal antibody conjugate therapy. This should clearly be given in the title, for instance by “Assessing the Clinical Effectiveness of Radioimmunotherapy with combined Radionuclide/Monoclonal Antibody Conjugates in Cancer Treatment: Insights from Randomised Clinical Trials”
Authors’ Response: Thank you for your valuable suggestion. We agree that the current title may not fully reflect the specific focus of our study on radionuclide/monoclonal antibody conjugate-based radioimmunotherapy. Accordingly, we have revised the title to:
“Assessing the Clinical Effectiveness of Radioimmunotherapy with Combined Radionuclide/Monoclonal Antibody Conjugates in Cancer Treatment: Insights from Randomised Clinical Trials.” Please see page 1.
Point 3: Simple summary an abstract section: Please harmonize number of patients 3562 or 3000?
Author’s response 3: Thank you. Please see page 3 of the abstract section where the ‘3000’ has been corrected to ‘3,562’.
Point 4: The comparative analysis in Table 3 indicates that radioimmunotherapy is predominantly integrated as part of combination regimens rather than given as mono therapy. Accordingly, treatment regimens given in table 3 should be included in table 2 to increase readability.
Author’s response 4: Thank you for highlighting this important issue regarding the presentation of treatment regimens. We have revised the manuscript as follows:
- We have merged the treatment regimen details from Table 3 into Table 2. The revised Table 2 now includes additional columns that specify whether radioimmunotherapy was administered as a monotherapy or as part of a combination regimen. This enhancement ensures that all relevant treatment information is consolidated in one place, making it easier for readers to quickly assess and compare the regimens across studies.
- The updated table format improves readability and facilitates a more comprehensive comparison, aligning with the objectives of our comparative analysis.
Point 5: Authors stressed the point that the monoclonal antibodies target tumor-associated antigens or those expressed by the tumor microenvironment, but failed to cover this information in the tables, e.g. table 1.
Author’s response 5: Thank you. Please see pages 13-15 of the Table 1 where a column of ‘antigen targeted’ has been added.
Point 6: It is not tractable to the reviewer, why authors indicate a number 20 studies to be included in the systematic analyses (figure 1) but displayed a total of 25 trials in table 2 and total of 20 studies in table 3. This discrepancy has to be solved.
Authors’ Response 6: We appreciate the reviewer’s attention to detail. The apparent discrepancy arises from the following points, which we have now clarified in the revised manuscript:
- Identification vs. Inclusion: In our initial search, a total of 25 trials were identified and summarised in Table 2. However, upon applying our inclusion/exclusion criteria and the quality assessment e.g., the Joanna Briggs Institute (JBI) Critical Appraisal Tool, only 20 studies met all the criteria for inclusion in the systematic analysis.
- Clarification in Figures and Tables:
- Figure 1: The flow diagram has been updated to clearly indicate that although 25 trials were initially identified, only 20 studies were ultimately included after applying the rigorous selection process.
- Table 3: Displays only the final 20 studies that passed the quality assessment and were included in the systematic analysis.
- Manuscript Revision: We have revised the relevant sections in the Methods and Results to ensure that the distinction between the initial pool of 25 trials and the final 20 studies is clearly explained, thereby resolving any confusion.
Round 2
Reviewer 1 Report
Comments and Suggestions for Authors
The raised comments have been adequately addressed in the revised version of the manuscript
Reviewer 4 Report
Comments and Suggestions for Authors
In the revised version of the manuscript, the authors addressed my previous concerns in an adequate manner.